# Preparation and Characterization of Bismaleimide-Resin-Based Composite Materials

**DOI:** 10.3390/ma17081727

**Published:** 2024-04-10

**Authors:** Lingrui Liang, Pei Wang, Zhihong Li, Yumei Zhu

**Affiliations:** Key Laboratory for Advanced Ceramics and Machining Technology of Ministry of Education, School of Materials Science and Engineering, Tianjin University, Tianjin 300072, China; lianglingrui@tju.edu.cn (L.L.); wpsun37@tju.edu.cn (P.W.); lizhihong@tju.edu.cn (Z.L.)

**Keywords:** composites, bismaleimide, silicon carbide, mechanical properties

## Abstract

This study utilized bismaleimide (BMI) resin, reinforced with introduced ether bonds, as a binding matrix, in combination with silicon carbide (SiC), for the fabrication of composite materials. A thorough investigation was conducted to assess the influence of diverse processing parameters on the mechanical properties and high-temperature thermo-oxidative stability of these composites. Experimental results indicate a notable improvement in the mechanical properties of the composites upon the incorporation of ether bonds, in contrast to their unmodified counterparts. The variation in performance among composites with different ratios and molding densities is apparent. Within a certain range, an increase in resin content and molding density is correlated with improved bending strength in the composites. With a resin content of 27.5 vol% and a molding density of 2.31 g/cm^3^, the composite achieved a maximum flexural strength of 109.52 MPa, representing a 24% increase compared to its pre-modification state. Even after exposure to high-temperature heat treatment, the composites displayed commendable mechanical properties compared to their pre-ether bond modification counterparts, maintaining 74.5% of the strength of the untreated composites at 300 °C. The scanning electron microscopy (SEM) microstructures of composite materials correlate remarkably well with their mechanical properties.

## 1. Introduction

Composite materials, characterized by their multiphase composition involving two or more components, possess distinctive material properties and find wide-ranging applications in contemporary technological advancements. They notably exhibit significant advantages in mechanical properties and high-temperature resistance, critical attributes for a plethora of high-performance applications [1,2]. Among these, silicon carbide (SiC)/resin composites stand out for their exceptional properties. Typically composed of silicon carbide particles or fibers embedded within a resin matrix, this composite material amalgamates the high-temperature stability and hardness of silicon carbide with the toughness and processability inherent to resin. Consequently, it exhibits outstanding capabilities in high-temperature environments and resistance to wear [3,4,5]. However, the resin component often faces challenges such as inadequate heat resistance and inferior mechanical properties, thereby imposing limitations on its application [6,7].

In recent years, there has been a concerted effort to modify resins in pursuit of improved properties. Bismaleimide (BMI) resin, recognized for its high-temperature resistance, has undergone rapid development. BMI resin is synthesized from two maleimide monomers via condensation reactions, resulting in a structure comprising two unsaturated maleimide groups. This unique configuration confers, upon BMI resin, exceptional thermal stability and mechanical properties. Curing involves the copolymerization of carbon-carbon double bonds, leading to the formation of a cross-linked network without the generation of small molecule by-products. This process results in minimal molding shrinkage, rendering BMI resin suitable for use as a high-temperature-resistant abrasive bonding agent. Consequently, it serves as an ideal matrix resin for high-performance composites and finds widespread application in aerospace, automotive, electrical and electronic, marine and oceanic, and petroleum chemical industries [8,9,10].

The increase in cross-link density in cured BMI materials often results in significant brittleness, posing a challenge to their practical application [11,12,13]. Additionally, unmodified BMI exhibits drawbacks such as a high melting point, low solubility, and a high molding temperature. The inherent brittleness, leading to diminished toughness, emerges as a key limitation impeding its development and utilization [14,15,16,17,18]. Overcoming this challenge while preserving their excellent heat resistance remains a pressing issue for BMI resins in practical applications. These inherent shortcomings collectively restrict the extensive use of BMI in high-performance engineering and advanced materials. Consequently, concerted efforts are required to address these limitations and broaden the scope of BMI resin applications [19,20,21].

In response to the challenges faced by BMI resin, various modification strategies have been explored to enhance its properties. Copolymerization modification using allyl compounds, such as Diallyl bisphenol A (DABPA), has emerged as a viable method for toughening BMI resins. This approach is considered one of the more mature toughening methods currently available [22,23]. The mechanism of copolymerization involves two key steps: the first is the opening of the carbon–carbon double bond in the BMI structure, leading to a diene addition reaction with allyl compounds to generate an intermediate conjugate. Subsequently, the intermediate conjugate undergoes a Diels–Alder reaction with BMI molecules, further forming a polymer cross-linking network and ultimately yielding a cured resin with a ladder-like structure [24,25]. Additionally, chain expansion modification represents another common approach to BMI molecular chain design modification [26,27,28]. This method primarily involves extending the main chain of BMI molecules through chemical reactions, introducing compounds with flexible chain segments such as low-molecular-weight compounds or polymers containing ether and ester bonds. This serves to increase the flexibility and mobility between molecular chains of the resin, thereby reducing the cross-link density of BMI and enhancing the toughness of its cured materials. Furthermore, the copolymerization of toughening modifications using allyl compounds and propylene-based compounds has been explored [29,30,31,32]. Allyl compounds, known for their mature modifier capabilities in BMI modification, introduce allyl-containing compounds that alter the molecular chain structure of BMI resin, thereby enhancing its flexibility and toughness through copolymerization or addition reactions with bismaleimide groups in the BMI resin system [33]. Alternatively, propylene-based compounds, boasting better reactivity than allyl compounds, can directly conjugate with the benzene ring, thereby reducing the modification process and lowering the curing reaction temperature of the modified resin. Furthermore, toughening modification through the addition of thermoplastic or thermosetting resins has been investigated [34,35,36,37]. This method involves altering the aggregation state structure of thermosetting resins by adding thermoplastic resin, forming a macroscopically homogeneous and microscopically two-phase microstructure. This effectively initiates crazes and shear ribbons, enabling the material to undergo significant deformation. Additionally, the hindering effect of thermoplastic resin particles on cracks prevents further crack expansion within the organization, thereby consuming more energy before destruction and achieving the purpose of toughening and modification [38]. Moreover, rubber or nano-materials with exceptional mechanical properties, such as carbon nanotubes and graphene, have been introduced to enhance the toughening and heat-resistant properties of BMI resin [39,40,41,42]. Rubber and BMI resin are typically formed through the hot–melt method or mechanical method of the homogeneous phase, leading to phase separation during the curing process and the formation of a “sea-island” structure. Rubber particles uniformly dispersed in the BMI matrix resin act as points of stress concentration, initiating a large number of crazes and shear ribbons upon impact, thus improving the resin’s impact strength [43]. On the other hand, nanoparticles serve as modified fillers of resin materials due to their small size and surface interface effects, enhancing the toughening and heat-resistant properties of organic resins. These nanoparticles not only absorb impact energy but also impart special properties to BMI cured materials [44].

In this study, the mechanical properties and heat resistance of SiC/modified resin composites were investigated using a 2,2-Bis(4-(4-Maleimidephenoxy)Phenyl) Propane (BMIX)-modified *N*,*N*′-(4,4′-methylenediphenyl) dimaleimide (BDM)/diallyl bisphenol A (DABPA) resin system.

## 2. Experiments

### 2.1. Materials

N, N′-(4,4′-methylenediphenyl) dimaleimide (BDM) (96%), 2,2-Bis(4-(4-Maleimidephenoxy)Phenyl)Propane(BMIX)(98%), and 4,4-(-(1-methylethylidene)bis [2-(2-propenyl)]phenol (DABPA) (90%) were supplied by Shanghai Bide Pharmaceutical Technology Co., Ltd. (Shanghai, China), 2,2′,2″-trihydroxytriethylamine (99%) was supplied by Zesheng Technology Co., Ltd. (Anqing, China), and silicon carbide (120 mesh) was supplied by White Pigeon (Group) Company Limited (Zhengzhou, China). The monomers and structures of some of the organic materials are shown in Table 1.

### 2.2. Experimental Procedureheng

Initially, DABPA and BMI/BMIX were precisely measured as starting materials, with a controlled ratio of BMI to BMIX set at n(BDM):n(BMIX) = 2:1. Utilizing the XU292 system as a reference, the BMI/BMIX and DABPA were blended in an oil bath at a ratio of n(BDM/BMIX):n(DABPA) = 1:0.87 at 130 °C, yielding prepolymers. Subsequently, the resulting products were naturally cooled, dried, crushed, and ground to obtain the final resin powder.

Subsequently, SiC particles and modified resin powder underwent screening through a 120-mesh sieve and were mixed according to predetermined proportions. The mixture underwent thorough homogenization and was sieved through the 120-mesh screen once more to ensure uniformity. Triethanolamine was added for wetting, and the mixture underwent further homogenization by passing through a 100-mesh screen. Following this, the mixture was cold-pressed, molded, and cured according to a predetermined curing temperature profile, resulting in the final cured BDM/DABPA/BMIX/SiC composite. Moreover, a control group was prepared by excluding BMIX from the resin powder mixture, maintaining the ratio of n(BDM): n(DABPA) = 1:0.87, and following the same procedural steps under identical environmental and experimental conditions.

### 2.3. Characterization

Fourier Transform Infrared Spectroscopy (FTIR): The characterization and analysis of uncured unmodified BMI resin and modified BMI resin systems were conducted using a Fourier Transform Infrared Spectrometer (FTIR, NicoletIS10, Thermo Fisher Scientific, Waltham, MA, USA) with a scanning range of 4000 to 400 cm^−1^ for systems post pre-polymerization at 130 °C.

Differential scanning calorimetry (DSC) analysis and thermogravimetric (TG) analysis: Thermal analysis of the prepolymers was carried out using a comprehensive thermal analyzer (DSC, TG, STA-449F3, NETZSCH Instruments Manufacturing GmbH, Selb, Germany) in an argon atmosphere with a ramp rate of 10 °C/min, spanning a temperature range of 25–1000 °C, and with a sample dosage of 5–10 mg.

Scanning electron microscopy (SEM) analysis and energy-dispersive spectroscopy (EDS): The fracture morphology of the composites subsequent to heat treatment and the bonding interface between SiC and resin were examined utilizing a field-emission scanning electron microscope (SEM, S-4800, Hitachi, Tokyo, Japan). Additionally, the distribution of the composites at the fracture was analyzed using energy-dispersive spectroscopy (EDS, X-MAX20, Oxford, UK).

Point bending tests were conducted at room temperature, utilizing a microcomputer-controlled electronic universal testing machine (CMT 4304, AT&S Industrial Systems Co., Ltd., Wuxi, China) with a deformation rate of 0.05 mm/s and specimen dimensions of (30 ± 0.2) mm × (6 ± 0.2) mm × (6 ± 0.2) mm. A rectangular specimen was subjected to a three-point bending test [45], and the flexural strength was calculated using Equation (1).
(1)$$\delta =\frac{3FL}{2bh^{2}}$$

Here,

*δ*—bending strength in megapascals (MPa);*F*—the maximum load that the specimen can withstand or that can reach the specified deflection value, in Newton (N);*L*—span between the two support points of the specimen, in millimeters (mm);*b*—width of the specimen in millimeters (mm);*h*—thickness of the specimen in millimeters (mm).

The thickness of the specimen was measured at three different points within the middle 1/3 of the specimen. The average thickness should not exceed 2% of the overall average thickness, and the width of the specimen should not exceed 3% of the average width of the specimen within this range.

## 3. Results

### 3.1. Modification and Structural Analysis

Throughout the curing process of BDM/DABPA resins and BDM/DABPA/BMIX resin systems, variations occur in the intensity of absorption peaks corresponding to each functional group as polymerization and chemical reactions progress. These variations arise from changes in the molecular structure of the polymer and the formation and disruption of chemical bonds among the functional groups. Fourier Transform Infrared Spectroscopy (FTIR) finds extensive application in characterizing the curing reactions among these functional groups. It enables the elucidation of vibrations associated with different functional groups in a sample, thus reflecting the progress of the chemical reaction. Figure 1 illustrates the FTIR spectra of the two polymer systems. By comparing their absorption peak positions and intensity alterations, valuable insights into the modifications of functional groups during the curing process can be obtained.

From Figure 1, it is evident that the absorption peaks observed at 1774 cm^−1^ and 1707 cm^−1^ correspond to the stretching vibration of imide C=O bonds while the absorption peak at 1637 cm^−1^ represents the variable angle vibration of water H-O-H. Additionally, the absorption peaks observed at 1607 cm^−1^, 1509 cm^−1^, and 1397 cm^−1^ signify the backbone vibration of benzene rings while those at 1263 cm^−1^ indicate the stretching vibration of phenol C-O bonds. Furthermore, the absorption peak at 1149 cm^−1^ corresponds to the stretching vibration of maleimide C-N-C bonds. The absorption peaks within the range of 1038 cm^−1^ to 900 cm^−1^ are attributed to the in-plane bending vibration of benzene-ring C-H bonds and the out-of-plane bending vibration of olefin C-H bonds. Lastly, the absorption peaks at 829 cm^−1^ signify the out-of-plane bending vibration of benzene-ring C-H bonds.

Following the introduction of BMIX, the vibrational groups exhibited similarity, resulting in minimal changes in the remaining absorption peaks. However, there were discernible shifts and alterations in the intensity of certain absorption peaks. Notably, the intensity of absorption peaks associated with the C=C group at 688 cm^−1^ and 915 cm^−1^ was attenuated, indicating a reduction in the characteristic peaks due to a double-bond addition reaction between BMI and DABPA. Moreover, a more prominent absorption peak at 1240 cm^−1^, corresponding to the stretching vibration of aromatic ether C-O-C bonds in BMIX, suggests the successful introduction of ether bonds into the polymer system.

### 3.2. Impact of Various Processing Parameters on Composite Mechanical Properties

The properties of composites are significantly influenced by the processing methods employed during their fabrication. Precise control over key process parameters is crucial to achieving desirable performance in the resultant composites. Concurrently, through the analysis and comparison of experimental outcomes, the disparities in composite properties under diverse process conditions can be comprehensively understood, thereby providing a theoretical framework for process optimization and performance enhancement. Therefore, it is imperative to meticulously regulate the preparation process and scrutinize its various compositional ratios, and additionally, the process parameters needed for use in subsequent DSC/TG studies must be determined.

The BDM/DABPA/BMIX resin powders obtained from previous experiments were mixed with SiC, keeping the resin binding agent content at 27.5 vol%, and the flexural properties of the composites were tested at molding densities of 1.76 g/cm^3^, 1.94 g/cm^3^, 2.13 g/cm^3^, 2.31 g/cm^3^, and 2.5 g/cm^3^ to investigate the influence of different molding densities on the mechanical properties of the composites. The experimental results are illustrated in Figure 2.

As illustrated in Figure 2, an increase in molding density corresponds to a rise in composite density. This phenomenon leads to a decrease in inter-particle spacing among abrasive grains, an expansion of the cross-sectional area, and a strengthening of bonding-agent bridges. Consequently, it enhances the adhesive strength between abrasive grains and the overall mechanical properties of the composites. Moreover, the elevated molding density increases the contact area between the resin and abrasive grains, facilitating improved adhesion between the abrasive grain surface and resin. This, in turn, reinforces the bending strength of the composite material. However, exceeding a molding density of 2.31 g/cm^3^ may result in a decline in strength. This outcome arises from the fact that excessive molding density can result in an elevated occurrence of defects within the material including heightened porosity. A comparison with the molding density depicted in Figure 3a reveals, in Figure 3b, the emergence of specific pores and cracks within the composite material due to excessive molding density. Additionally, an excessively dense or non-uniform internal arrangement of the composite material can lead to incomplete or uneven curing, resulting in structural defects such as voids, cracks, and other deformations. Ultimately, this compromises the mechanical properties of the composite.

Therefore, adjusting the molding density of SiC/resin composites constitutes a critical approach to modulating their mechanical properties. This is aimed at optimizing composite density, minimizing particle spacing, and reinforcing bonding-agent bridges, thereby improving mechanical performance.

The resin content, recognized as a critical parameter in composite fabrication, plays a significant role in determining the mechanical properties of the resulting material. In this study, experiments were conducted using pre-prepared BDM/DABPA/BMIX resin matrices combined with SiC, with resin contents ranging from 20 vol% to 30 vol%. The molding process maintained a consistent density of 2.31 g/cm^3^. The primary objective was to investigate the impact of resin content on the bending strength of the composite materials. The experimental results are depicted in Figure 4.

Figure 4 reveals a discernible trend in the bending properties of the composites, characterized by an initial increase followed by a subsequent decrease as the content of modified BMI resin escalates. With the content of modified BMI resin escalating from 20 vol% to 27.5 vol%, there is a linear increase in the bending property from 37.29 MPa to 109.52 MPa. Nevertheless, as the content of modified BMI resin is further elevated to 30 vol%, the bending property of the composite material declines to 52.43 MPa. An examination of the scanning electron microscope images depicted in Figure 5 indicates that the observed phenomenon may be ascribed to the inadequate filling effect at lower resin concentrations. Specifically, as illustrated in Figure 5a, in comparison to the scenario depicted in Figure 5b wherein the resin content is 27.5%, at a resin content of 20%, the resin fails to entirely fill the voids, thereby leading to a decrease in the contact area between the resin and the silicon carbide particles. With an increase in resin content, the voids in the composites are filled, reducing porosity. Concurrently, the resin can establish a robust bond with the surface of SiC particles, augmenting particle bonding force, overall density, and hardness, consequently ameliorating the bending properties of the composites. However, when the resin content exceeds 27.5 vol%, further increases reduce the bending performance of the composite material. This could be due to excessive resin content leading to excessive bonding between the resin and abrasive particles, resulting in the formation of bulky or large resin structures that hinder effective particle bonding. Additionally, excessive resin may cause issues such as foaming, gap formation, or deformation in the composite material, with the presence of defects being a primary reason for the reduction in bending strength.

### 3.3. Mechanical Properties and Thermal Stability Analysis of Composite Materials

Figure 6 illustrates the DSC analysis curves of BDM/DABPA resin and BDM/DABPA/BMIX resin. The melting and heat absorption of BDM/DABPA/BMIX resin occur around 90 °C. The decrease in BMI’s melting point not only enhances its processing performance but also decreases the density of reactive groups per unit volume in the resin system. Consequently, it lowers the cross-linking density of the modified BMI-cured resin, thereby further enhancing its toughness. Subsequent analysis revealed that the BDM/DABPA/BMIX resin system exhibits two prominent exothermic peaks. The initial peak, occurring at approximately 155.4 °C, can be interpreted as showing an ongoing Alder-Ene reaction between the C=C bond of the allyl group in DABPA and the C=C bond within the BMI molecule’s ring structure. This phase of reaction can be construed as the pre-polymer not being fully reacted but rather continuing the pre-polymerization process, thereby fostering further elongation of the molecular chain and enhancing material toughness. The subsequent exothermic peak, observed at approximately 257.5 °C, may be attributed to the ongoing self-polymerization and Diels–Alder addition reactions among BMI molecules, alongside intermolecular cross-linking curing processes.

The glass transition temperature (T_g_) holds paramount significance in polymer performance, denoting a sudden shift in numerous physical attributes, notably mechanical properties. The glass transition temperature (T_g_), ascertained via the inflection point method, is defined as the temperature corresponding to the peak value of the differential scanning calorimetry (DSC) signal or the apex of the slope within the transition region. In the BDM/DABPA/BMIX resin system, the T_g_ was measured at 274 °C. This T_g_ value suggests that the composites retain superb mechanical properties even at elevated temperatures of 250 °C, hence exhibiting potential for prolonged usage.

Further comparative analysis revealed that the thermal characteristics of the BDM/DABPA/BMIX resin system demonstrated commendable stability under elevated temperatures. According to the data in Figure 7 and Table 2, the system exhibits a T_5_ of 409 °C, a T_15_ of 432 °C, and a T_30_ of 463 °C, which do not deviate significantly from those of the BDM/DABPA resin system. Although the overall decomposition of the BDM/DABPA/BMIX resin system was slightly lower, and the mass retention rate at 1000 °C was 35.4%, slightly lower than the 38.9% of the BDM/DABPA resin system, indicating a certain degree of thermal stability at high temperatures, it also suggests a slight decline in thermal stability and a weakening of decomposition resistance. Nevertheless, it is noteworthy that these temperatures lie beyond the conventional operating temperature range of the resin. Consequently, the introduction of the ether bond did not exert a substantial impact on the thermal stability of the resin system within the operational temperature range. Additionally, through a comparison of the mechanical property data in Figure 8, it is evident that the BDM/DABPA/BMIX/SiC composites continue to manifest superior properties compared to the BDM/DABPA/SiC composites post high-temperature treatment.

The specimens of BDM/DABPA/BMIX/SiC composites and BMI/DABPA/SiC composites were placed in an air atmosphere furnace and heat-treated at constant temperatures of 270 °C, 300 °C, 350 °C, 400 °C, and 450 °C for 1 h. Subsequently, their flexural strength was tested after the high-temperature treatment. In this study, the BDM/DABPA/BMIX/SiC composites were chosen with a molding density of 2.31 g/cm^3^ and a resin content of 27.5 vol%. Likewise, the BMI/DABPA/SiC composites were fabricated under identical conditions.

The changes in the flexural strength of the resin composites before and after modification, as well as the influence of temperature on their flexural strength, were investigated through a comparison of the mechanical properties of modified BMI resin/SiC composites with conventional BDM/DABPA/SiC composites after high-temperature treatment. At room temperature, the mechanical properties of BDM/DABPA/BMIX/SiC composites significantly exceed those of BDM/DABPA/SiC composites. This enhancement can be attributed to the introduction of ether bonds, which enhance molecular flexibility and decrease the cross-linking density of the cured resin, thereby improving its toughness and overall mechanical properties. The effects of temperature on the flexural strength of the composites are depicted in Figure 8, revealing a gradual decrease in properties after 1 h heat treatment at different temperatures. Particularly noteworthy is that BMI/DABPA/BMIX/SiC composites exhibit only a slight decrease at 270 °C, followed by a more pronounced decline between 270 °C and 300 °C. This sudden decrease may be attributed to the glass transition temperature of the BMI/DABPA/BMIX resin system being reached, leading to an abrupt reduction in properties. Subsequently, between 300 °C and 400 °C, the strength of BMI/DABPA/BMIX/SiC composites decreases at a more uniform rate, indicating structural damage to the BMI resin system and the partial decomposition of the maleimide, as evidenced by the DSC graph in Figure 6 and the thermogravimetric graph in Figure 7.

### 3.4. Composite Fracture Surface Analysis

An examination of the microstructures of the fractured composite material surface was conducted via scanning electron microscopy, with corresponding test results illustrated in Figure 9 and Figure 10. Figure 9 depicts the scanning electron microscopy image of the fracture surface of the BDM/DABPA resin system material incorporating silicon carbide, whereas Figure 10 exhibits the scanning electron microscopy image of the fracture surface of the BDM/DABPA/BMIX resin system material with silicon carbide following the introduction of ether bond modification. The morphology of SiC particles encompasses prisms, hexagonal columns, flakes, and other forms, presenting a smooth angular shape. Moreover, the microstructure of the resin matrix predominantly manifests a relatively smooth surface.

Figure 9a,b depict the SEM images of the cross-sections of BDM/DABPA/SiC composite materials pre- and post-1 h heat treatment at 270 °C, respectively. It is evident that, at this juncture, the SiC particles within the composite material exhibit robust bonding with the resin matrix, encapsulated within it devoid of any cracks or pores. With the elevation of the heat treatment temperature to 300 °C, microcracks begin to emerge within the resin matrix, as depicted in Figure 9c. Subsequent increments in heat treatment temperature to 350 °C and 400 °C, as illustrated in Figure 9d,e, result in the appearance of pores and substantial cracks within the resin matrix, accompanied by partial exposure of SiC particles. Upon reaching 450 °C, Figure 9f showcases the resin matrix adorned with extensive cracks and minute pores, rendering it incapable of effectively bonding with SiC particles. Consequently, the composite material loses its mechanical properties, corroborating the findings of the three-point bending test.

Figure 10 presents the microstructure of the cross-section of the BDM/DABPA/BMIX/SiC composite material. Comparing Figure 9a and Figure 10a, the introduction of BMIX brings more toughness fracture to the composite material compared to before modification. The SEM images depicted in Figure 10a,b illustrate the cross-sections of the composite material prior to and following 1 h heat treatment at 270 °C, respectively. At this stage, the adhesion between the resin and SiC particles is robust. Upon reaching a heat treatment temperature of 300 °C, small bubbles become apparent in Figure 10c, with the resin still firmly bonded to SiC without visible cracks. However, with an increase in the heat treatment temperature to 350 °C, voids and bubbles emerge in the composite material cross-section, as depicted in Figure 10d, accompanied by indications of incomplete encapsulation of silicon carbide particles by the resin, resulting in particle exposure. This phenomenon may stem from the rupture of specific chemical bonds within the resin matrix. In Figure 10e, the composite material cross-section post-1 h heat treatment at 400 °C is displayed. At this juncture, the composite material strength diminishes to only 22.4% of its initial strength. Evident voids and significant cracks at the interface junction indicate substantial damage to the resin structure, with some resin fragmented and detached. Upon reaching a heat treatment temperature of 450 °C, the resin structure undergoes severe damage, resulting in adhesive failure between silicon carbide particles. Pronounced cracks emerge at the SiC particle–resin junction, with mechanical strength nearing 0 megapascals, consistent with the mechanical performance results obtained via three-point fracture testing.

The uniform mixing of components within composite materials contributes to enhancing overall structural equilibrium and stability. An EDS spectrometer (Figure 11) was employed to analyze the constituents within the bright white regions present in the cracks and matrix separately. Silicon (Si) and nitrogen (N) elements were specifically chosen to assess material dispersion, with silicon originating from SiC and nitrogen from the aromatic ring in BMI resin. The distribution of these elements can influence mechanical properties. Analysis revealed a relatively uniform distribution of these elements. Figure 11 illustrates that SiC particles neither agglomerate nor distribute unevenly, with the resin effectively encapsulating SiC particles. Post-curing, the composite material maintains homogeneity without aggregation. The favorable distribution of Si and N elements, coupled with the uniform dispersion of SiC particles within the matrix, augments the mechanical properties of resin-based composite materials.

## 4. Conclusions

This experiment involved the preparation and characterization of composite materials through the incorporation of modified BMI with SiC. When the molding density is 2.31 g/cm^3^ and the resin content is 27.5 vol%, the bending performance reaches its maximum value. Compared with the unmodified BMI/SiC composite material, the mechanical properties of the modified BMI/SiC composite material are significantly improved. This improvement was evident upon comprehensive evaluation encompassing mechanical properties, thermal analysis, and SEM scanning images while thermal stability was concurrently maintained.

Various process parameters exert a profound influence on the mechanical properties of composite materials. Notably, differing molding densities significantly impact the overall mechanical reinforcement of these materials. Insufficient molding density may result in inadequate material density, consequently diminishing its mechanical resistance, whereas excessive molding density can induce an uneven or excessively dense internal arrangement of composite materials, leading to incomplete curing or uneven distribution, thus impairing mechanical properties. Hence, optimal resin content plays a pivotal role in enhancing the overall mechanical properties of composite materials while excessive resin content may induce deformation and pore formation, contributing to a decline in mechanical properties.The glass transition temperature (Tg) of the modified BMI system exceeds 270 °C, with the temperature at 5% weight loss surpassing 400 °C. Moreover, the maximum degradation rate temperature is lower than that observed before modification, suggesting minimal reduction in the heat resistance of the BMI system following the introduction of ether bonds.The incorporation of ether bonds led to a substantial enhancement in the mechanical properties of BDM/DAPPA/BMIX/SiC composites. Without high-temperature heat treatment, the modified BMI/SiC composites attained a bending strength of 109.52 MPa, marking a 24% increase compared to their pre-modification counterparts. Furthermore, the microstructure of the composite material, devoid of high-temperature heat treatment, exhibited negligible voids or cracks, underscoring the robust bonding between the resin and silicon carbide particles at typical operating temperatures, ensuring prolonged utility.A comparative analysis of BDM/DAPPA/SiC composite materials and BDM/DAPBA/BMIX/SiC composite materials following high-temperature heat treatment revealed the superior performance of the former. High-temperature heat treatment at 270 °C, 300 °C, 350 °C, and 400 °C resulted in enhanced mechanical properties compared to unmodified composite materials. Notably, at 270 °C and 350 °C, the bending strengths were 98.28 MPa and 65.69 MPa, respectively, representing 24% and 26% increases over unmodified composite materials. Furthermore, the thermal stability of the modified composite material exhibited minimal deterioration post-modification, ensuring the retention of satisfactory mechanical properties.

## Figures and Tables

**Figure 1 materials-17-01727-f001:**
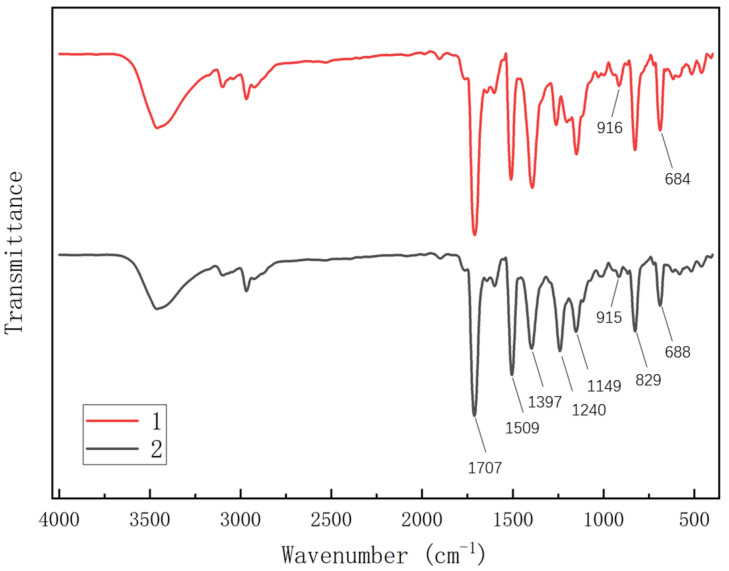
Fourier Transform Infrared (FTIR) spectra of polymer systems after pre-polymerization: (**1**) BDM/DABPA; (**2**) BDM/DABPA/BMIX.

**Figure 2 materials-17-01727-f002:**
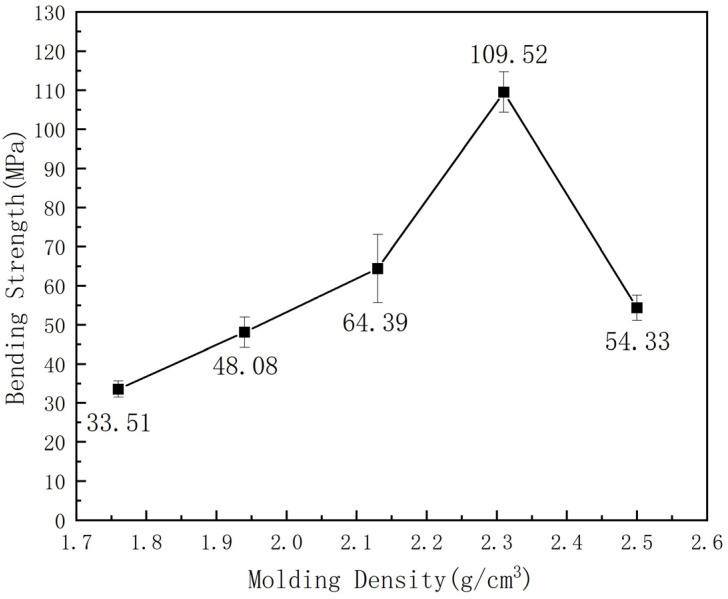
Bending strengths of composites with different molding densities.

**Figure 3 materials-17-01727-f003:**
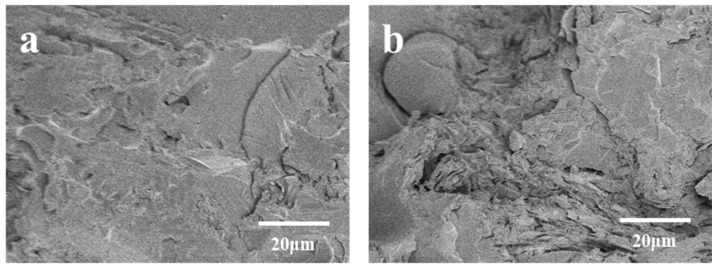
SEM images of the cross-sections of composites with different molding densities: (**a**) 2.31 g/cm^3^; (**b**) 2.5 g/cm^3^.

**Figure 4 materials-17-01727-f004:**
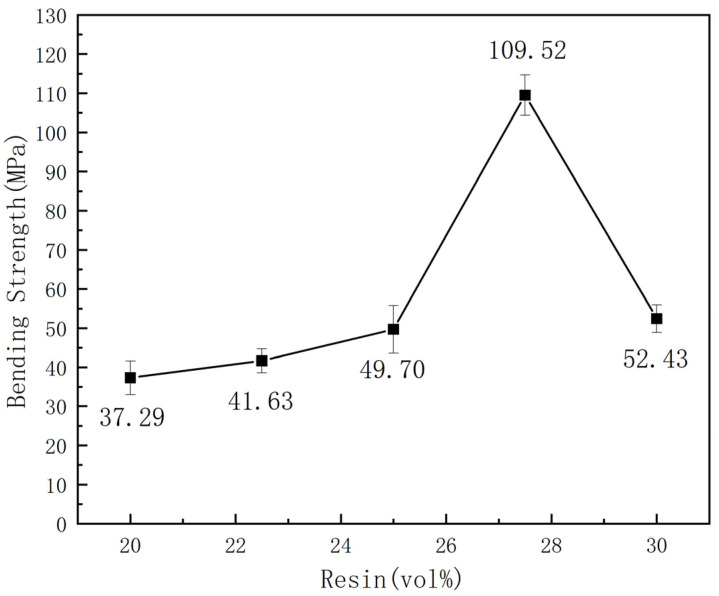
Bending strengths of composites with different resin contents.

**Figure 5 materials-17-01727-f005:**
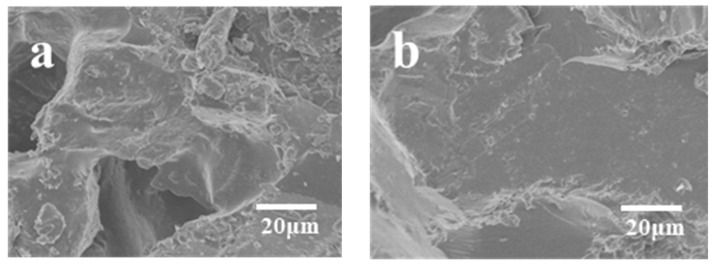
SEM images of the cross-sections of composites with different resin contents: (**a**) 20 vol%; (**b**) 27.5 vol%.

**Figure 6 materials-17-01727-f006:**
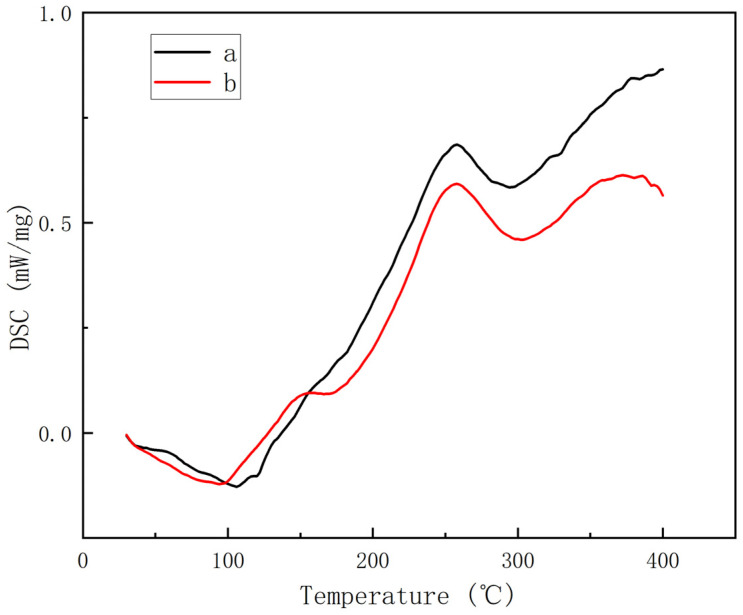
DSC curves of resin after pre-polymerization at 130 °C: (**a**) BDM/DABPA; (**b**) BDM/DABPA/BMIX.

**Figure 7 materials-17-01727-f007:**
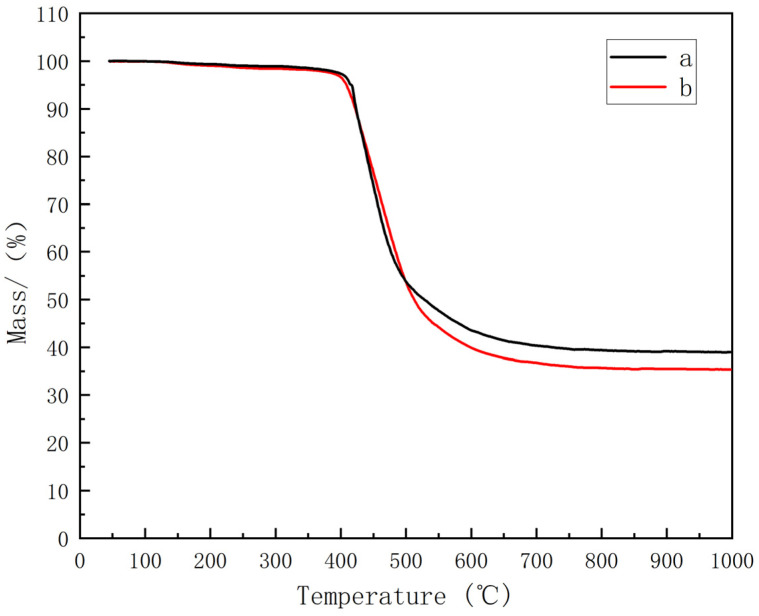
Thermogravimetric curve of resin: (**a**) BDM/DABPA; (**b**) BDM/DABPA/BMIX.

**Figure 8 materials-17-01727-f008:**
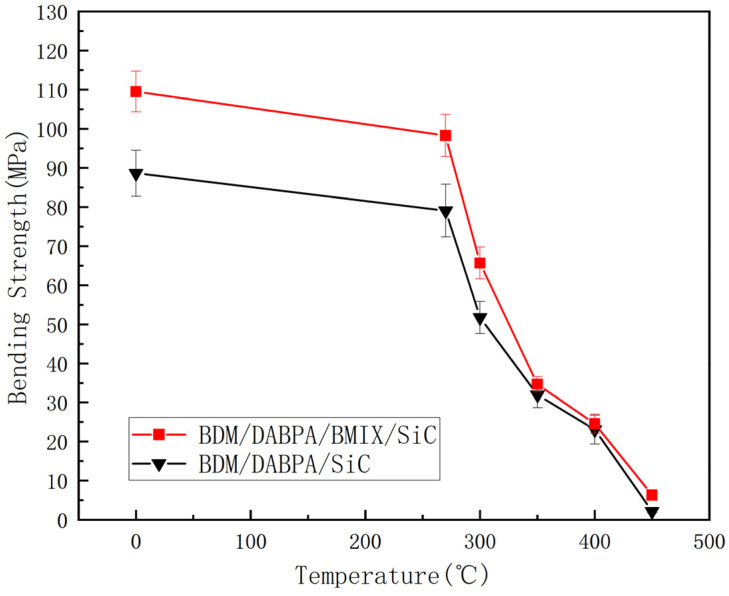
Effect of temperature on the bending strengths of composites.

**Figure 9 materials-17-01727-f009:**
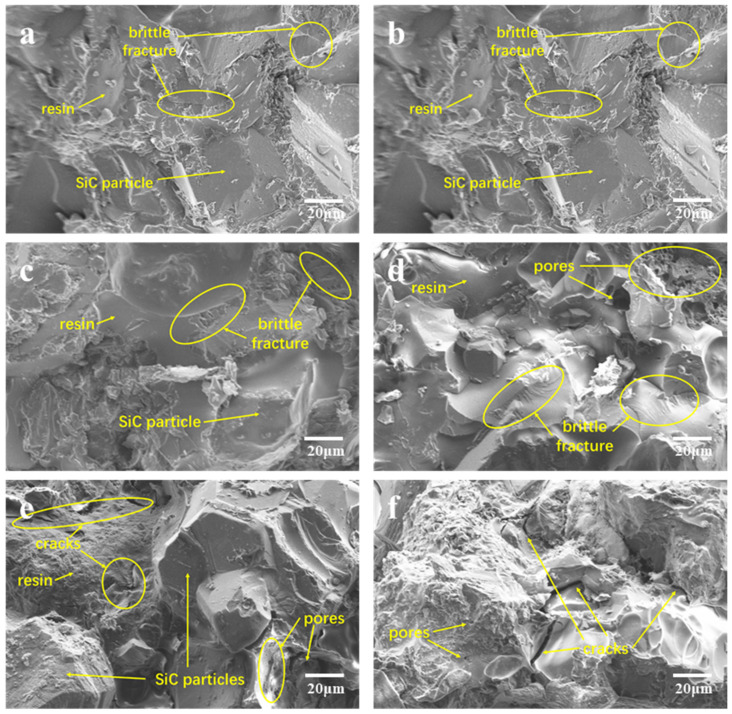
SEM micrographs of fracture surfaces of BDM/DABPA/SiC composites after heat treatment at different temperatures: (**a**) not heat-treated; (**b**) 270 °C; (**c**) 300 °C; (**d**) 350 °C; (**e**) 400 °C; (**f**) 450 °C.

**Figure 10 materials-17-01727-f010:**
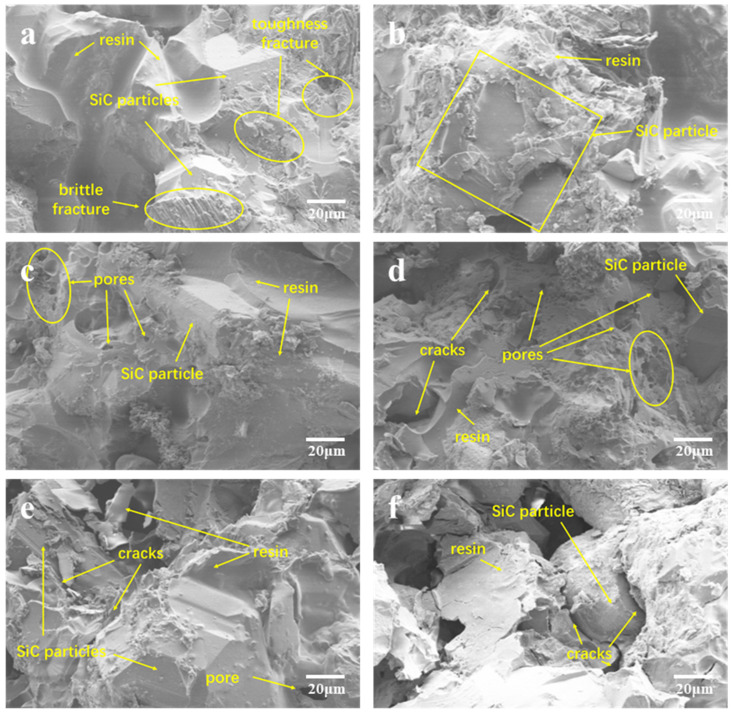
SEM micrographs of fracture surfaces of BDM/DABPA/BMIX/SiC composites after heat treatment at different temperatures: (**a**) not heat-treated; (**b**) 270 °C; (**c**) 300 °C; (**d**) 350 °C; (**e**) 400 °C; (**f**) 450 °C.

**Figure 11 materials-17-01727-f011:**
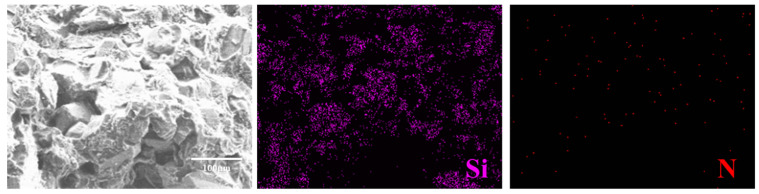
EDS spectra of fracture surfaces of BDM/DABPA/BMIX/SiC composites.

**Table 1 materials-17-01727-t001:** Chemical structures of tested materials.

Monomers	
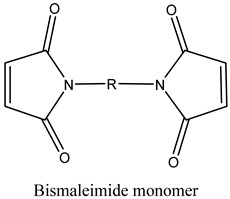	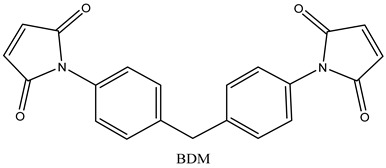
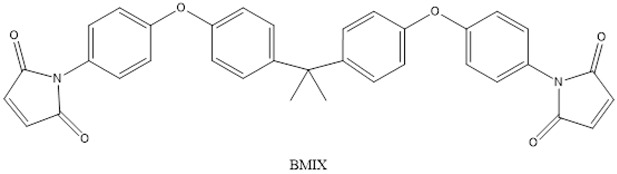	
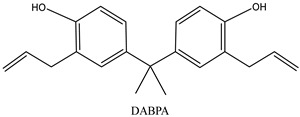	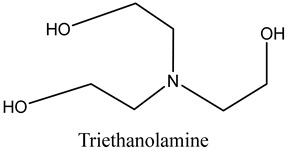

**Table 2 materials-17-01727-t002:** Resin heat resistance TG results.

Samples	Weight Loss Temperature (°C)			Maximum Degradation Rate Temperature (°C)
	5%	15%	30%	
BDM/DABPA	416	431	456	420
BDM/DABPA/BMIX	409	432	463	469

## Data Availability

Data are contained in the article.

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
