# Peer review of "Preparation and Characterization of Bismaleimide-Resin-Based Composite Materials"

_materials, 2024, doi:10.3390/ma17081727_

Round 1

Reviewer 1 Report

Comments and Suggestions for Authors

The manuscript, titled “Preparation and Characterization of Bismaleimide Resin-Based Composite Materials”, submitted to MDPI Materials by Laing et al. presents results on the synthesis and characterisation of silicon carbide-bismaleide phenolic resin composites. The resins were synthesised from 4,4-(-(1-methylethylidene)bis[2-(2-propenyl)]phenol (DABPA) and N, N'-(4,4'-methylenediphenyl) dimaleimide (BDM) / 2,2-Bis(4-(4-Maleimide-phenoxy)Phenyl)Propane(BMIX) and were characterised by means of FTIR spectroscopy, differential scanning calorimetry (DSC) and thermogravimetric analysis (TGA), as well as via SEM/EDS. The effects on preparation parameters, such as moulding density and composition, such as resin content on the mechanical properties of the resins were investigated, by means of point-bending tests. 

In the results - the authors conveyed an optimization survey on the optimal moulding density and resin content to yield the highest bending strength of the BDM/DABPA/BMIX resin, which were found to be 2.31 g/cm3 and 27.5 vol.%, respectively and then TGA and DSC analysis was conducted on samples, produced at these conditions to elucidate the thermal behaviour of the resin/SiC composites. The main conclusions are that the addition of BMIX provides enhanced mechanical properties and improved thermal resistance, due to better binding with the inorganic filler.

In general I am satisfied with the manuscript and believe that it would be of interest to the readers of MDPI Materials. The manuscript has a clearly stated research idea and seems scientifically sound. The quality of the English presentation is good and the text is easy to follow, even though I should note that the overabundance of superlatives (such as “meticulously”, “profound”, “imperative”, “commendable”, etc.) makes me feel like I could speculate that certain language-model probably had some assistance in the final corrections (which if true, is not against the MDPI guidelines).

All in all, I am in favour for the publication of the manuscript and would only suggest some minor corrections, which are listed below:

(1) I suggest that the wavenumbers are added in the FTIR spectra in Figure 1, below the transmittance minima of the main peaks, discussed in the text. This would ease its use by a potential reader seeking for FTIR references. On a minor note - the correct notation on Fig. 1 y-axis is “Transmittance” and not “Transmission” /* and additionally - Wavenumber should probably be singular.

(2) In subsection 3.2., it is not initially clear that the purpose was to optimise the preparation parameters of the model system used in the TGA/DSC study and probably should be stated more clearly. 

(3) In subsection 3.3. it is not immediately clear that the results discussed there are for the composite prepared at 2.31 g/cm3 moulding density and 27.5 vol.% resin content and it should be underlined to help the reader. I also suggest that this information is noted in figure & table captions in this subsection.

(4) The “Conclusions” section should be improved - it is quite general in this states, verging, on sounding auto-generated. It should include more details about the findings in the main text and mention more numerical data - currently only the optimal parameters for resin preparation are mentioned there.

Reviewer 2 Report

Comments and Suggestions for Authors

Congratulation, it is a great work. The manuscript follows the improvement of mechanical behavior of bismaleimide resin-based composite by: improving polymer matrix binding by introducing ether bonds and using different filler/matrix ratio. SEM images are excellent and describe well the samples microstructure and the 3 point flexural test was well conducted. The polymer matrix improvements were characterized by FTIR and DSC correlated with TG. The experimental setup is well organized based on literature data study sustained by proper references. The conclusions are sustained by the obtained results. There are some aspects that require revisions. Please follow the comments below:

Comment 1) Lines 89 – 90 the term ,,triggers silver lines” is very ambiguous and require clarifications or rephrasing. The same situation is for the line 99.

Comment 2) Lines 98 -99: usually rubber forms soft microstructural features that do not act as stress concentrators. Therefore, the phrase needs clarifications and maybe some references regarding this behavior.

Comment 3) It is supposed that in Tables 2, 3 and Figures 2, 3, 7 are presented the bending strength mean values (of at least 3 determinations), in consequence the standard deviation ± must introduced beside the values in the tables and as error bars on the experimental points of the figures.

Comment 4) Lines 345 – 349 are the repetition of lines 341 – 345, the repeated sentences must be removed.

Comment 5) Microstructural aspects observed in Figures 8 and 9 must be presented in a more detailed manner in text and each image must be discussed and cited individually (for example Figure 8f presents some cracks with the local filler particles de-laminations due to the bending failure...)

Comment 6) An example of SEM fractography of a composite failure characterization is found at the following DOI: https://doi.org/10.3390/biomedicines11071965

Comment 7) Figure 10 – The elemental map of Si is excellent revealing the filler particles distribution but the elemental map of N is completely blind there cannot be observed anything. The graphical quality of N elemental map must be improved (try to increase brightness and contrast). Also try to expand discussion about elemental distribution effect on the failure mode.

Comment 8) Lines 377 – 378 ,,This enhances the reinforcing effect of the reinforcing phase and improves the mechanical properties of the composites.” must be rephrased.

Comments on the Quality of English Language

Minor editing of English language required

Reviewer 3 Report

Comments and Suggestions for Authors

This study present the enhancement of mechanical properties and thermal stability of Bismaleimide Resin-Based Composite Materials with incorporation of ether bonds into bismaleimide resin, combined with silicon carbide (SiC) reinforcement. I recommend this paper for publication, with minor revisions to clarify some experimental approaches and provide additional details on the SEM analysis. Here are my comments:

1.      The details of the heat treatment in the Experiment section are unclear. Could you elaborate the conducted treatment procedure?

2.      In Table 2, the results of moulding density on bending strength have been reported. The authors claim that exceeding of molding density higher than 2.31 g/cm3 may result in a decline in bending strength. The reason of decline in the paper has attributed to porosity. However, there is no evidence in the paper that shows or measures the porosity.

3.      Table 3 and Fig.3 state same results. It is recommended to remove Table3 and instead label the bending strength values at each respective point on the graph.

4.      Can the authors show inadequate filling effects in the SEM picture (Fig.4)?

5.      Line, 327, there is a typing error, the effects of temperature on the flexural strength of the composites are depicted in Figure 6. Figure7 is correct.

6.      The authors should characterize, firstly, the BDM, DABPA, SiC, and BMIX in the SEM pictures. Secondly, they should explain the effect of the heat treatment and BMIX on the fracture type.

7.      In conclusions, No.2, The authors mentioned “the microstructure of the composites exhibits no discernible holes or cracks, indicating a robust bond between the resin and SiC particles”. However, I can see some holes and cracks in the SEM picture (for ex. Fig.9 f), so we cannot say that there is a robust bond between the resin and SiC particles
